# A Comparison of Common Quantum Dot Alternatives to Cadmium-Based Quantum Dots on the Basis of Liver Cytotoxicity

**DOI:** 10.3390/nano14131086

**Published:** 2024-06-25

**Authors:** Seth Harris, Kyoungtae Kim

**Affiliations:** Department of Biology, Missouri State University, 901 S National, Springfield, MO 65897, USA; seth0018@live.missouristate.edu

**Keywords:** quantum dots, fluorescent nanoparticles, CdSe/ZnS, InP/ZnS, CuInS_2_/ZnS, NCD, HepG2, THLE-2, liver cytotoxicity

## Abstract

Fluorescent nanoparticles known as quantum dots (QDs) have unique properties that make them useful in biomedicine. Specifically, CdSe/ZnS QDs, while good at fluorescing, show toxicity. Due to this, safer alternatives have been developed. This study uses a tetrazolium dye (XTT) viability assay, reactive oxygen species (ROS) fluorescent imaging, and apoptosis to investigate the effect of QD alternatives InP/ZnS, CuInS_2_/ZnS, and nitrogen-doped carbon dots (NCDs) in liver cells. The liver is a possible destination for the accumulation of QDs, making it an appropriate model for testing. A cancerous liver cell line known as HepG2 and an immortalized liver cell line known as THLE-2 were used. At a nanomolar range of 10–150, HepG2 cells demonstrated no reduced cell viability after 24 h. The XTT viability assay demonstrated that CdSe/ZnS and CuInS_2_/ZnS show reduced cell viability in THLE-2 cells with concentrations between 50 and 150 nM. Furthermore, CdSe/ZnS- and CuInS_2_/ZnS-treated THLE-2 cells generated ROS as early as 6 h after treatment and elevated apoptosis after 24 h. To further corroborate our results, apoptosis assays revealed an increased percentage of cells in the early stages of apoptosis for CdSe/ZnS-treated (52%) and CuInS_2_/ZnS-treated (38%) THLE-2. RNA transcriptomics revealed heavy downregulation of cell adhesion pathways such as wnt, cadherin, and integrin in all QDs except NCDs. In conclusion, NCDs show the least toxicity toward these two liver cell lines. While demonstrating less toxicity than CdSe/ZnS, the metallic QDs (InP/ZnS and CuInS_2_/ZnS) still demonstrate potential concerns in liver cells. This study serves to explore the toxicity of QD alternatives and better understand their cellular interactions.

## 1. Introduction

The field of biomedicine is always looking for new tools for improving imaging, detection, diagnosis, drug delivery, and others. An emerging tool to help with many of these aspects could be quantum dots. These tiny nanoparticles (2–10 nm) found in many electronics such as LED televisions, phones, and LED lights have gained attention not only as a research tool, but also in biomedical applications [1]. Materials science has seen the adoption of many nanomaterials such as metal oxides, carbon nanotubes, and quantum dots for use in biomedicine [2]. Recently, the alteration of nanomaterials for biomedical use has gained attention because of their unique properties.

Quantum dots are semiconductor nanocrystals that can be synthesized from a variety of materials such as Cd, Pb, In, C, Si, and polymers [3]. Specifically, quantum dots (QDs) are of interest because they contain the ability to fluoresce when excited and can be tuned to specific wavelengths [3]. While QDs have many applications in electronics, they also have been modified to be more biocompatible and suitable for biomedical applications [4]. Cancer-targeted therapy and bioimaging are two applications for quantum dots that have potential [4].

While there is immense potential, their applications are currently limited to in vivo studies because of possible toxicity [5]. Cadmium-based QDs are very effective in producing high quantum yield; however, research has shown them to be quite toxic [4]. The makeup of cadmium quantum dots is heavy-metal-based, making toxicity a concern. This concern also extends to the environment with the increased industrial use of quantum dots [4]. Many different cell lines have been used to showcase the toxic effects of cadmium-based quantum dots, including cervical cancer (HeLa), thyroid cancer (ML-1), and liver cancer (HepG2) [6,7,8]. Over the years, safer alternatives such as indium-based and carbon-based quantum dots have been developed. These alternatives are newer and need more research attention. This is especially true for carbon-based quantum dots (C-dots) as their non-metallic base could prove to be the safest option for a core. A recent study found no effect of carbon quantum dots on zebrafish development [9]. Metal-based QDs like cadmium can demonstrate toxicity if these heavy metals leak out of the core [10]. To help prevent this, zinc sulfide (ZnS) shells have been added around the quantum dot cores to improve stability [10]. In conjunction with ZnS shells, surface modifications are added to quantum dots in order to increase their biocompatibility [11]. While compatibility increases, this could also make the quantum dots react negatively within cells. As surface modifications commonly added to QDs, COOH and NH_2_ can interfere with cellular trafficking as well as other cellular pathways [11]. Another example of damaging interactions includes the generation of reactive oxygen species (ROS), which can damage cell structure, by quantum dots [12]. Quantum dot interactions in yeast, specifically interactions of cadmium-based QDs, have also been studied and shown to interact with cytoskeletal structure and disrupt actin polymerization [13]. To add to the knowledge of QD toxicity, we explored their safety in two liver cell lines known as HepG2 and THLE-2.

We tested the cellular effects of four QDs, namely CdSe/ZnS (cadmium-based), InP/ZnS (indium-based), CuInS_2_/ZnS (indium-based), and nitrogen-doped carbon dots (NCDs, carbon-based), in liver cells. As QD size can play a role in toxicity by affecting uptake, the QDs used in this study were carefully selected to be of similar sizes all ranging from 6.5 to 9 nm to reduce differences in this variable. Uniquely, no other study directly compares QD alternatives to their toxic CdSe/ZnS counterpart. We chose an under-utilized cell line (THLE-2) because it maintains a human cell line model while being proliferative and noncancerous. We hypothesized that the InP/ZnS, CuInS_2_/ZnS, and NCD alternatives will prove to be safer options. Most notably, we expect NCDs to have the least effect on both cell lines. Broadly, this study serves to compare cell viability, ROS generation, and apoptosis among CdSe/ZnS QD alternatives in an immortalized liver cell line (THLE-2) that acts as healthy liver cells.

## 2. Materials and Methods

### 2.1. Quantum Dots

All quantum dots were sourced from NN-labs (Fayetteville, AR, USA). CdSe/ZnS quantum dots (Product #HECZW450-5) fluoresce at 450 ± 10 nm and are 7–8 nm in diameter. InP/ZnS quantum dots (Product #HEINPW590-5) fluoresce at 590 ± 15 nm and are 6.5–8.5 nm in diameter. CuInS_2_/ZnS quantum dots (Product #CISW750-5) fluoresce at 750 ± 15 nm and are 6.6 ± 0.5 nm in diameter. All three of these QDs were solubilized in 5 mg/5 mL of water. InP/ZnS QDs were specifically diluted to a stock concentration of 1.0 µM to match the other two quantum dots. In addition, all three QDs contain a carboxylic acid stabilizing ligand and ZnS shell that protects and further stabilizes the quantum dot core structure. The carboxylic acid ligand and ZnS shell serve to improve the overall stability and biocompatibility of the QDs. The carbon dots were sourced from Dr. Wanekaya’s lab at Missouri State University. They are specifically nitrogen-doped carbon dots (NCDs) with a size of 8–9 nm in diameter. The stock concentration of these carbon dots is 1.5 µM.

### 2.2. Cell Lines

We used two liver cell lines in this study. Many models have been used to test quantum dot interactions with biological organisms, including mammalian cells, cancerous cells, yeast, and plants [4]. Several studies suggest that QDs will accumulate in the liver, so they are a relevant model for testing the toxicology of QDs [14,15,16]. One cell line, HepG2, is a cancerous hepatocellular carcinoma cell line. While this cell line is cancerous, it is a popular model for toxicology as the cells maintain normal function for metabolizing drugs [17]. The other, THLE-2, is an immortalized liver cell line. THLE-2 is considered immortalized because the simian virus 40 large T antigen gene was introduced into human adult liver epithelial cells to allow the cell to continuously proliferate like a cancer cell [18]. The cell line demonstrated nontumorigenic properties in mice, so it can be used as a model for normal, healthy liver cells [18]. There have been studies testing quantum dots in HepG2, but few studies using THLE-2 as a liver cell model.

### 2.3. Cell Culture

We grew both cell lines in 25 cm^2^ flasks with 6 mL of their respective media. The cell culture was grown in a 37 °C incubator with 5% CO_2_. HepG2 was cultured in RPMI-1640 medium containing 25 mM HEPES and L-Glutamine (Cytiva, MA, USA) and THLE-2 using the Lonza Walkersville BEGM Bullet Kit (CC-3170, MD) (Lonza, Basel, Switzerland) containing bronchial epithelial cell basal medium (BEBM) with the following growth supplements: BPE, 2 mL; Hydrocortisone, 0.5 mL; hEGF, 0.5 mL; Epinephrine, 0.5 mL; Transferrin, 0.5 mL; Insulin, 0.5 mL; Retinoic Acid, 0.5 mL; Triiodothyronine, 0.5 mL; GA-1000, 0.5 mL. In addition, both media were enhanced with 10% fetal bovine serum (FBS) and 1% antibiotics (penicillin and streptomycin). We grew cells until confluent and used them for seeding and treatment of QDs. To detach the cells from the flask, we added 1.5 mL of trypsin with EDTA after the removal of media and multiple 1xPBS washes. The trypsin was neutralized using 6 mL of fresh media and vigorous pipetting up and down. Next, the cells were centrifuged at 400 rpm for 10 min to form a cell pellet. We resuspended the pellet with 6 mL of fresh media. Quantification of cell number was conducted using 10 µL of cells with a hemocytometer.

### 2.4. QD Characterization

#### 2.4.1. XPS Chemical Composition

X-ray photoelectron spectroscopy (XPS) was used to verify the chemical composition of each of the QDs used in this study. Sample preparation for XPS was the same for all samples, where 10 µL was pipetted onto clean silicon (Si) substrates and allowed to dry in air in a desiccant dry box. XPS measurements were carried out using a Thermo Scientific Nexsa System equipped with a microfocused, monochromated, Al Kα (1486.6 eV) X-ray radiation source. All spectra were collected using a 400 µm spot size in a ~8 × 10^8^ mbar vacuum environment. The survey scans were collected at a pass energy of 200 eV and binding energy (BE) steps of 0.10 eV. All data analysis was conducted using Thermo Scientific AVANTAGE v5 software and compared to several XPS databases (https://srdata.nist.gov/xps/ (accessed on 2 June 2024), https://xpsdatabase.com/ (accessed on 2 June 2024), https://www.xpsfitting.com/ (accessed on 2 June 2024). All peaks were charge-compensated to the adventitious C peak at 284.5 eV.

#### 2.4.2. Fluorescent Spectra Analysis

We prepared a sample concentration of 50 and 100 nM for each QD. Subsequently, we measured the innate fluorescence of each quantum dot. For these measurements, we used a PTI spectrofluorometer (PTI Photon Technology International, Birmingham, NJ, USA). We set an excitation wavelength at 365 nm and an emission range of 400–900 nm. In addition, we set excitation bandwidths of 8 nm and emission bandwidths of 10 nm. All data sets were graphed using GraphPad Prism 9.

### 2.5. XTT Viability Assay

We used an XTT kit from Biotium to test the viability of QD-treated liver cells. We first detached the cells and seeded them into a 96-well plate. Due to the size of the cells, we seeded HepG2 at a density of 10,000 cells/well and THLE-2 at a density of 8000 cells/well. We grew the cells overnight in their respective media and treated them the next day. Using a serial dilution, we treated the cells with a variety of QD concentrations. We aliquoted the following concentrations and applied 100 µL of QD/media solution: 150 nM, 100 nM, 50 nM, and 10 nM. We included a non-treated control, and each treatment was conducted in triplicate. After treatment, we incubated the cells for 24 h in a 37 °C incubator with 5% CO_2_. On the third day, the XTT kit (Biotium, Fremont, CA, USA) protocol was followed using the manufacturer’s instructions. We created the XTT reagent using a combination of tetrazolium salt XTT and an activating solution in a 200:1 ratio. To each well, we added 25 µL of XTT reagent, and we incubated the wells at 37 °C with 5% CO_2_ for 5 h. The XTT salt (tetrazolium) is reduced by mitochondrial enzymes to provide an orange product. This assay is colorimetric and can provide absorbance measurements that correspond to levels of cell viability using spectrophotometry. We read the measurements at the delta wavelength range (450–630 nm) using a BioTek ELx880 (Agilent, Santa Clara, CA, USA) plate reader (VT). Finally, we analyzed our data using GraphPad Prism9 software (CA) to find *p* values < 0.05 and calculate IC20 values. The following formula was used to calculate the IC20 value: ${IC}_{f}={\left(\frac{100-F}{F}\right)}^{1/H}\times {IC}_{50}$. The variables came from the Hill slope in GraphPad Prism9 [19]. The value F is the percentage of maximal inhibition (20%), and H is the Hill slope which was given by GraphPad Prism 9 after statistical analyses of the results.

### 2.6. ROS Assay

First, we seeded cells into an ibidi 96-well black glass bottom plate (catalog #89627, WI) (ibidi, Munich, Germany) with 100 µL of their respective media. The plate was coated with 0.1% collagen according to the manufacturer’s directions for improved cell adherence. The seeding density was the same as that for the XTT procedure (8000 cells/well for THLE-2). On the second day, we treated the cells at either IC20 values or a baseline concentration. The IC20 values were as follows: CdSe/ZnS, 61 nM; CuInS_2_/ZnS, 84 nM. The other two treatments, InP/ZnS and C-dots, were treated at a baseline value of 100 nM. Again, treatments were conducted in triplicate. In one experiment, we incubated cells with QDs in a 37 °C incubator with 5% CO_2_ for 6 h; in another, incubation lasted 24 h. Immediately after incubation, 100 µL of 5 µM DCF from Thermo Fischer Scientific, Waltham, MA, USA (H2DCFDA, catalog #D399) was added to each well. We incubated the cells with DCF in a 37 °C incubator for 30 min to allow the dye to bind ROS. After incubation, the culture with DCF was removed, and we washed the wells very carefully with 1×PBS. We kept the cells in 1×PBS and imaged them at 20× using confocal fluorescence microscopy.

#### Confocal Microscopy

Three images were captured per well, containing roughly 20 cells per image and totaling 60 cells. Images were captured in triplicate, creating a sample size of 180 cells total per treatment. The average fluorescent intensity was evaluated using ImageJ v1.54 software and entered into GraphPad Prism9 for statistical analysis. In ImageJ, a square gate of 25 × 25 pixels was used to measure the fluorescent intensity of each cell.

### 2.7. Apoptosis Assay 

To investigate the levels of apoptosis after QD, we used an apoptosis kit from BD Biosciences, Franklin Lakes, NJ, USA. First, we cultured THLE-2 cells and seeded them at a cell density of 50,000 cells/well in a 24-well plate. We treated wells in triplicate for each QD and used IC20 concentrations or 100 nM (for QDs with no IC20). After 24 h, we collected the cells using trypsin with no EDTA (Corning, Corning, NY, USA) and added apoptotic dyes. We added 5 µL of propidium iodide (PI) and Annexin V-APC to each well. The wells also contained 100 µL 1× Annexin V binding buffer. After 30 min of incubating in the dark, an additional 400 µL of buffer was added, and the levels of apoptosis were measured using flow cytometry. Annexin V-APC is a marker for early apoptosis, and propidium iodide is a marker for late apoptosis.

#### Flow Cytometry Gating

We created three blanks, one containing no dye and the other two with one of each of the dyes. These were used as compensation to set the gates for apoptosis. Using a polygon gate, we selected all cells for each blank and left out any high-density background noise. Using a density plot, we selected a quadrant gate and placed it at the center of the different blank populations. This provided the right location to compare non-treated controls to treated cells. We used an Attune NxT acoustic flow cytometer (Life Technologies-Thermo Fisher Scientific, Waltham, MA, USA). Annexin V-APC had an excitation/emission filter set at 650/660 nm. PI’s filter was set at 533/616 nm.

### 2.8. RNA Transcriptomics

#### 2.8.1. RNA Isolation

We seeded THLE-2 cells at a density of 500,000 cells/well in a 6-well plate. The following day, we treated them in triplicate at IC20 values or 100 nM if the IC20 was not applicable. After 24 h, we isolated the RNA. For isolation, we added 1 mL of TRIzol reagent to each well and incubated the wells at room temperature for 3 min to lyse the cells. Following cell lysis, we combined the lysate with 200 µL chloroform in 1.5 mL microcentrifuge tubes. After shaking the tubes to mix their contents, we centrifuged the samples at 12,000 rcf (4 °C) for 15 min. We transferred the aqueous phase into a new tube along with 500 µL of isopropanol. We incubated this tube at 4 °C for 10 min. Then, we centrifuged the samples for 10 min at 12,000 rcf (4 °C). After pellet formation, we performed resuspension in 1 mL 75% ethanol. Another brief centrifugation step was conducted for 5 min at 7500 rcf (4 °C). We removed the supernatant and allowed the remaining ethanol to evaporate. We resuspended the pellet in RNase-free water for use in further applications or storage (−70 °C).

We then assessed the quality of the RNA using the Agilent TapeStation 4150. We combined 1 µL of RNA sample with 5 µL RNA sample buffer into a tube strip. Then, we mixed the samples using an IKA MS3 vortexer (IKA, Wilmington, NC, USA) and centrifuged the samples at 2000 rpm for 1 min. Immediately after this, we placed the tube strip into a Veriti Thermo Cycler for 3 min at 72 °C. After heating, we placed the tube strip on ice for 2 min. One more round of centrifugation was applied before the TapeStation was used. We used the TapeStation controller software v5.1 to find the RNA integrity numbers (RINs). Values > 8 were accepted.

#### 2.8.2. RNA Seq Analysis

Quality RNA was shipped to Novogene (Sacramento, CA, USA). Novogene conducted directional mRNA library preparation. The raw transcriptomic data were then concatenated using Galaxy V24.1 (usegalaxy.org) and imported into CLC Genomics Workbench 24 (QIAGEN, Germantown, MD, USA) to be trimmed and gathered into a gene list. The reference genome used was GRCh38. After acquiring the gene list, we sorted the differentially expressed genes based on upregulation (fold change > 1.5, *p*-value < 0.05) and downregulation (fold change < −1.5, *p*-value < 0.05).

Gene list analysis was conducted using the Panther 18.0 knowledgebase to sort up- and downregulated genes into pathways. We then graphed the pathway data using GraphPad Prism 9. Alternatively, models were created using the ShinyGo 0.80 gene enrichment tool developed by South Dakota State University (bioinformatics.sdstate.edu/go, accessed on 20 March 2024) and KEGG (Kyoto Encyclopedia of Genes and Genomes). The gene list was set to show the top 50 pathways with an FDR cutoff of 0.05. All models were created using BioRender.com.

### 2.9. Statistical Analysis

Statistical analysis using GraphPad Prism9 was used for all assays. Statistical analysis consisted of entering data into a column analysis and then using a one-way ANOVA test to find statistical differences and Dunnett’s test for multiple comparisons. A *p*-value of <0.05 was considered significant and notated with an asterisk on all figures. The *p*-value of 0.05 is represented by a single asterisk and decreases to 0.01, 0.001, and 0.0001 with subsequent asterisks. XTT viability assay variables for calculating IC20 values were gathered using nonlinear regression using a dose–response inhibition equation in GraphPad Prism9.

## 3. Results and Discussion

### 3.1. XPS Component Analysis

The X-ray photoelectron spectroscopy (XPS) spectra for each QD (CdSe/ZnS, CuInS2/ZnS, InP/ZnS, and NCD) fit the database data for peak position, indicating chemical state and verifying chemical composition. Figure 1A–D provide survey scans for each measured QD, showing qualitative elemental composition. Figure 2, Figure 3, Figure 4 and Figure 5 provide the high-resolution XPS spectra for each QD constituent alone and the corresponding peak spectra table with a comparison to established database values. The high-resolution spectra for each QD show the associated peaks, along with their peak positions. The elemental peak positions were compared to the database-published values for the associated chemical state, which agrees with the collected XPS measurements.

### 3.2. QD Peak Fluorescence

Each quantum dot naturally fluoresces at different wavelengths. While size usually determines the wavelength of fluorescence, we attempted to keep the size of each QD consistent. In this case, the differing wavelengths are due to the QD composition. Using a fluorometer, we were able to find the fluorescent spectra for each QD after exciting them at 365 nm. Both CdSe/ZnS and NCDs peak around 450 nm in the blue wavelength. InP/ZnS peaks around 600 nm in the orange wavelength. Finally, CuInS_2_/ZnS fluoresces around 750 nm in the red wavelength (Figure 6). Each of these values aligns with the manufacturer. Notably, the peak fluorescence is concentration-dependent as each QD has a different quantum yield. CdSe/ZnS and InP/ZnS peaked as shown using only 50 nM, while CuInS_2_/ZnS and NCDs peaked using 100 nM. According to the manufacturer (nn-labs), the quantum yield for CdSe/ZnS is >50%, InP/ZnS is >30%, and CuInS_2_/ZnS is >15%. A paper published on our nitrogen-doped carbon dots in 2023 revealed their quantum yield to be ~20% [20]. We made sure to perform treatments at different concentrations to better represent the difference in quantum yield. CdSe/ZnS quantum dots are treated at 61 nM, CuInS_2_/ZnS at 84 nM, and InP/ZnS and NCDs at 100 nM. One limitation is that CuInS_2_/ZnS could not be treated any higher due to toxicity.

### 3.3. CdSe/ZnS and CuInS_2_/ZnS Show Reduction in THLE-2 Cell Viability

We determined the cytotoxicity of each treatment using an XTT assay. The results show that in THLE-2, CdSe/ZnS and CuInS_2_/ZnS QDs reduce cell viability in a dose-dependent manner beginning at 100 nM for CdSe/ZnS (Figure 7A) and 50 nM for CuInS_2_/ZnS (Figure 7C). The IC20 values for CdSe/ZnS and CuInS_2_/ZnS were calculated to be 61 nM and 84 nM respectively. This is the concentration in which there is a 20% reduction in cell viability. The other CdSe/ZnS alternatives, InP/ZnS and NCDs, do not affect cell viability at the concentrations tested (Figure 7B,D). In addition, the THLE-2 cell line is more sensitive to quantum dots as none of the treatments reduced cell viability in HepG2 cells (Figure 7E–H).

To start, we found that CdSe/ZnS QDs and CuInS_2_/ZnS QDs can reduce cell viability in THLE-2 cells. An in vivo study using BALB/c mice found that PEGylated CuInS_2_/ZnS did not demonstrate toxicity to organs after 90 days post-injection, hinting at the importance of surface modifications in QD toxicity [21]. Considering their CuInS_2_/ZnS QDs were PEGylated, this could add to further stability and reduce Cu^2+^ leakage/toxicity. Similar CuInS_2_/ZnS QDs with a carboxylic ligand (COOH) treated in *Gobiocyprus rarus* (minnow) embryos reported many developmental defects [22]. In THLE-2 cells, it takes very low concentrations of CdSe/ZnS (100 nM) and CuInS_2_/ZnS (50 nM) to show adverse effects on cell viability (Figure 1A,C). In corroboration with our viability results in HepG2, a study testing similar concentrations of CdSe/ZnS QDs (10–100 nM) revealed no toxicity [8]. This study used CdSe/ZnS with a PEG coating which our QDs did not have. Regardless, the CdSe/ZnS QDs demonstrated no toxicity in HepG2. Based on our cell viability assay alone, InP/ZnS and NCDs show potential as safer quantum dot alternatives. Specifically, nitrogen-doped carbon dots have been tested in other models and reveal low toxicity. A study tested the toxicity of nitrogen-doped carbon dots in both HeLa cells and Swiss albino mice [23]. This study found no toxicity when looking at physiological parameters in mice and no evidence of DNA fragmentation or apoptosis in HeLa cells [23].

### 3.4. CdSe/ZnS- and CuInS_2_/ZnS-Treated THLE-2 Cells Exhibit Early ROS Generation

To further determine the toxic effects of these four QDs on THLE-2 cells, we investigated the generation of reactive oxygen species (ROS). ROS generation was only investigated in THLE-2 cells for their relevance as a noncancerous, human cell line model and the results of XTT. We incubated THLE-2 cells with each quantum dot treatment for 6 and 24 h. The measured fluorescent intensity corresponds to ROS generation based on DCF binding. After 6 h, fluorescent intensity significantly increased in all treatments except carbon dots. CdSe/ZnS had the greatest increase in fluorescent intensity (Figure 8C). CuInS_2_/ZnS had the second greatest increase followed by InP/ZnS (Figure 8C). Again, NCDs caused no significant increase in fluorescent intensity compared to the NTC after 6 h (Figure 8C). After 24 h, all treatments were shown to elevate ROS with a very highly statistically significant difference with a *p*-value < 0.0001 (Figure 8D).

In line with the XTT results, CdSe/ZnS and CuInS_2_/ZnS show a significant elevation of ROS after 6 h which could explain the loss in cell viability. After 24 h, all QDs demonstrated a significant elevation of ROS. For InP/ZnS and carbon dots, the generation occurred slowly, which could explain why their XTT results show more cell viability. The literature around ROS generation by quantum dots is divisive due to the sensitivity of the process. ROS can fluctuate, which may lead to variability in results, making this a potential limitation. A study testing CdSe/ZnS and InP/ZnS in HeLa and ML-1 cells concluded that the reduction in cell viability was not due to oxidative stress [7]. A separate study looking at CdSe/ZnS in HeLa cells concluded the same [6]. However, in HFF-1 cells, CdSe/ZnS QDs with a carboxylic acid ligand caused elevated ROS, specifically mitochondrial ROS, between 4 and 6 h after treatment, which agrees with our results [11]. 

### 3.5. CdSe/ZnS and CuInS_2_/ZnS Treatment Causes THLE-2 Cells to Enter Early Stages of Apoptosis

When investigating apoptosis after a 24 h QD treatment, we noticed differences in activation of the early stages of apoptosis. This is made evident by a clear shift in quadrants observed in the flow cytometry plots (Figure 9A). Namely, CdSe/ZnS induced an average of 52% cells bound with Annexin V-APC, and CuInS_2_/ZnS an average of 38% (Figure 9B). When compared to the NTC, we observed InP/ZnS and NCDs to have no discernible changes as around 16% of the cells demonstrated the marker for early apoptosis (Figure 9B).

When exploring the induction of early- and late-stage apoptosis, we found similar trends in toxicity to those outlined by the cell viability and ROS assays. CdSe/ZnS and CuInS_2_/ZnS evidently induce apoptosis even at 61 and 84 nanomolar concentrations. It is important to note that early induction of apoptosis does not prove full completion of apoptosis; however, it is evidence for the favoring of death mechanisms even with low amounts of QDs. InP/ZnS and NCDs even at higher concentrations of 100 nM do not exhibit this induction of apoptosis (Figure 9B). In hepatic L02 cells, another study found the induction of cell death by CdSe/ZnS-treated cells by pyroptosis, an inflammatory-mediated cell death mechanism [24]. In HepG2, one paper found no effect on cell viability or cell death in CuInS_2_/ZnS-treated cells [25]. In cancerous cells, CuInS_2_/ZnS treatment reveals little cytotoxicity. However, in THLE-2, the present study demonstrates a more sensitive response to QDs that may be better representative of the response of healthy liver cells. Early induction of apoptosis marked by the Annexin V-APC dye is characterized by the movement of phosphatidyl serine (PS) to the outer leaflet of the cell membrane which can occur in multiple types of cell death [26]. The present study showcases that CdSe/ZnS and CuInS_2_/ZnS-treated THLE-2 cells are beginning cell death but does not specify the exact cell death mechanism, which is a limitation of the assay used.

### 3.6. Transcriptomics Reveals Heavy Downregulation of Adherence Pathways

When treating cells with each QD for 24 h, we were able to detect a number of upregulated and downregulated genes. Most notably, CdSe/ZnS- and InP/ZnS-treated cells exhibited the most downregulation, with around 1000 genes (Table 1). For upregulation, CuInS_2_/ZnS-treated cells had the strongest effect, nearing 200 genes. Comparatively, NCDs had the fewest changes in the transcriptome, with fewer than 50 genes being up- or downregulated (Table 1). GO enrichment via the Panther database (pantherdb.org) revealed significant downregulation of adhesion-related pathways for all QDs except the carbon dots (NCDs) (Figure 10). Specifically, the pathways of interest were as follows: wnt, cadherin, and integrin. Most genes downregulated were involved in various levels of metabolism; however, no clear distinction could be drawn between these genes and the four QDs tested. To avoid arbitrarily picking metabolic pathways to investigate, the pathways chosen were affected the most without being related to metabolism. CdSe/ZnS and InP/ZnS shared the most similar profile downregulating wnt signaling with 26 and 23 genes, respectively (Figure 10). Similarly, cadherin pathways were downregulated with 21 and 16 genes, respectively (Figure 10). Finally, integrin pathways were downregulated with 12 genes for both CdSe/ZnS and InP/ZnS. CuInS_2_/ZnS demonstrated less downregulation overall but still demonstrated downregulation of integrin pathways primarily with six genes affected (Figure 10). NCDs did not present this defect in adhesion pathways in THLE-2 cells.

### 3.7. Wnt, Cadherin, and Integrin Pathways Are Downregulated by Metallic QDs

We discovered a common trend in the downregulation of the cadherin and integrin adherence pathways as well as the closely related wnt pathway. Specifically, CdSe/ZnS and InP/ZnS favor the downregulation of wnt and cadherin signaling while CuInS_2_/ZnS favor the downregulation of the integrin pathway. The specific genes of interest include Wnt11 (wnt family member 11) and CDH15 (cadherin 15). CdSe/ZnS-treated THLE-2 cells had a fold change of −18.83 for wnt11 and −90.23 for CDH15. InP/ZnS demonstrated fold changes of −7.11 for wnt11 and −10.38 for CDH15. In addition, the false discovery rate (FDR)-adjusted *p*-value fell below 0.05.

A common trend we found in metallic-based QD treatments was the ability to downregulate genes related to cell structure and adhesion. Wnt signaling is a diverse pathway with implications in cell proliferation, cell polarity, differentiation, and cell motility [27]. Both the canonical and noncanonical (PCP) wnt pathways demonstrated significant downregulation (Figure 11). The downstream effects of this dysregulation could lead to cytoskeletal defects, particularly with actin polymerization [28]. In yeast models, research has been conducted demonstrating the direct inhibition of actin polymerization by CdSe/ZnS quantum dots [29]. Our study indicates an additional indirect mechanism that could be affecting actin polymerization, notably in a human cell line. Cadherins play a crucial role in establishing adherens junctions that keep cells tightly together. Loss of cell-to-cell adhesion could lead to abnormal tissue morphogenesis [30]. In addition, collagen genes related to integrin and the extracellular matrix were significantly downregulated, pointing to more changes in structurally related genes. The changes to integrin pathway expression are the primary downregulated adhesion pathway in CuInS_2_/ZnS-treated THLE-2 cells favoring this over cadherin and wnt signaling (Figure 11). The minimal downregulation caused by NCD treatment resulted in no clear pattern of genetic alteration.

### 3.8. CD82, SYT12, and DUSP4 Are Upregulated by Metallic QDs 

When analyzing patterns of upregulation, we found few common genes between QDs. Only three shared genes were evident between the metallic QDs. CD82 (a tetraspanin superfamily glycoprotein), synaptogamin 12 (SYT12, related to the SNARE complex and vesicle trafficking), and Dual specificity protein phosphatase 4 (DUSP4, which is involved in reducing oxidative stress). CD82 has many diverse implications including cell death through ER stress, autophagy, or oxidative stress [31,32,33]. In addition, overexpression of CD82 is linked to the downregulation of wnt signaling and integrins related to cell motility [34]. The gene DUSP4 codes for an antioxidant, which reveals the attempt of the CdSe/ZnS-, InP/ZnS-, and CuInS_2_/ZnS-treated THLE-2 cells to reduce the generation of oxidative stress [35]. The success of this survival mechanism varied among treatments, as shown by the varying levels of ROS and apoptosis revealed in earlier figures (Figure 4 and Figure 5). The upregulation of SYT12 may provide insight into the possible trafficking of QDs out of the cell. SYT12 is typically associated with neuronal cells [36], so the upregulation in THLE-2 liver cells is unexpected. However, all three metallic QDs (CdSe/ZnS, InP/ZnS, and CuInS_2_/ZnS) upregulated this gene in THLE-2, which is why it was reported (Table 2). SYT12 is a non-calcium-dependent synaptogamin that facilitates spontaneous neurotransmitter release [36]. There is little literature on the role of SYT12 in the liver, so further investigation would be needed to classify the role of this gene in response to QD treatment. NCD-treated THLE-2 cells revealed no clear pattern of upregulation or similarities to the other QDs.

## 4. Future Directions

In the future, we will look to further categorize the downregulation of adherence-related genes and continue to explore the downstream consequences caused by QDs. In addition, we will further highlight the cellular defects shown by investigating the cell morphology by confocal microscopy and including protein-based assays to support the transcriptomic data.

## 5. Conclusions

Considering quantum dots’ diverse range of applications, it is important to understand how they interact with living organisms. The development of safer, cadmium-free alternatives creates the need for researching their cellular effects, especially when considering biomedical applications. In this study, we compare the liver cytotoxicity of common cadmium-free quantum dots to CdSe/ZnS QDs. This screening of QD alternatives most notably elucidates the potential for carbon QDs to be sufficient replacements for cadmium-based counterparts as there were no significant effects on cell viability, apoptosis, or the transcriptome in THLE-2 liver cells. InP/ZnS can also be considered a safer alternative if kept at low concentrations. The transcriptomic profile of InP/ZnS was very similar to that of CdSe/ZnS even though the levels of ROS and apoptosis were lower (Figure 4 and Figure 5). Essentially, while less severe toxicity is indicated, there is still concern about whether higher dosages of InP/ZnS are needed to reach the same fluorescent capacity as CdSe/ZnS. CuInS_2_/ZnS, however, demonstrates liver toxicity on par with CdSe/ZnS, and more understanding of its toxicity mechanism is required.

## Figures and Tables

**Figure 1 nanomaterials-14-01086-f001:**
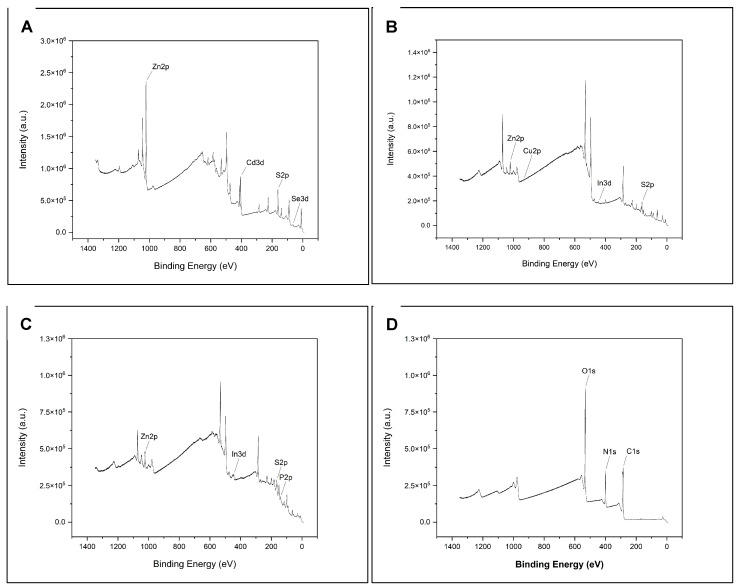
XPS spectra for each quantum dot. The graphs show the binding energy and peak intensity for each component of the QD. (**A**–**D**) represent a specific QD as labeled.

**Figure 2 nanomaterials-14-01086-f002:**
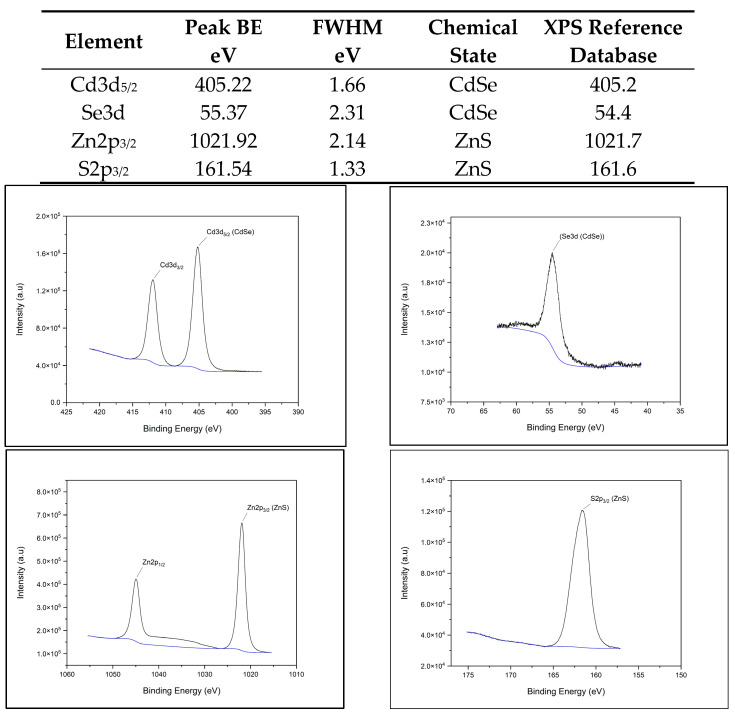
CdSe/ZnS QD high-resolution XPS spectra with corresponding peak table.

**Figure 3 nanomaterials-14-01086-f003:**
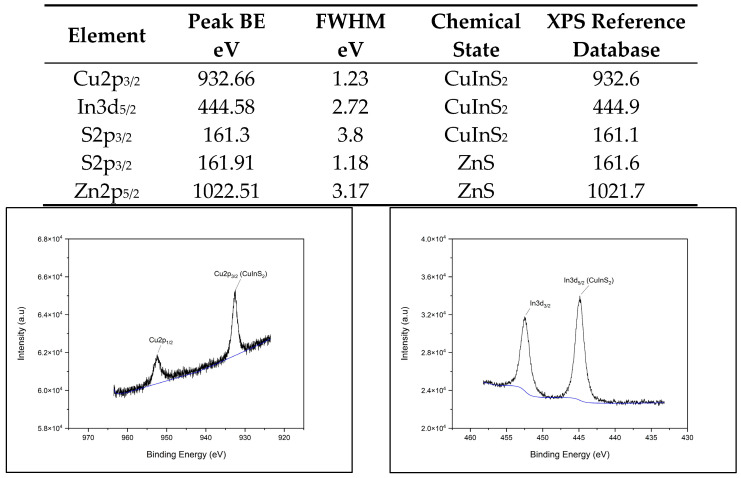
CuInS2/ZnS QD high-resolution XPS spectra with corresponding peak table.

**Figure 4 nanomaterials-14-01086-f004:**
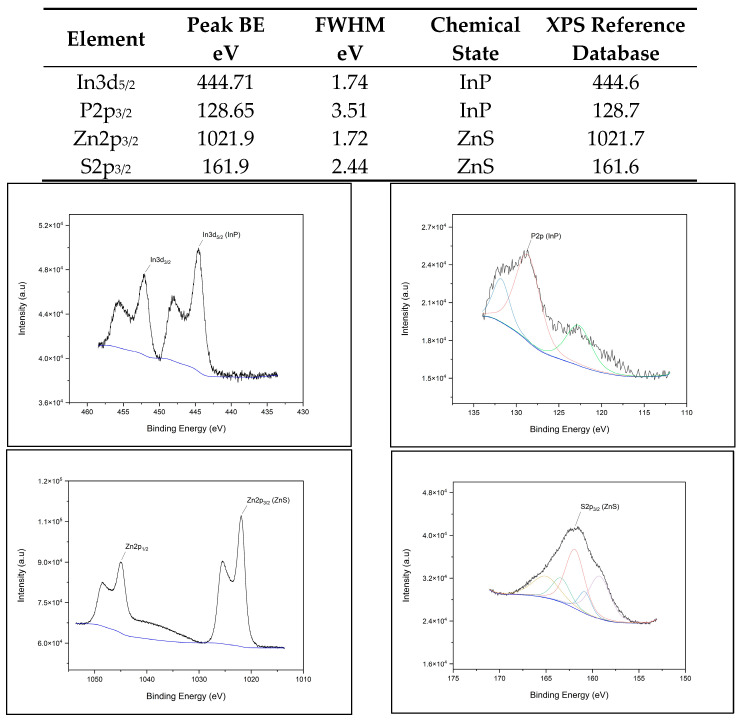
InP/ZnS QD high-resolution XPS spectra with corresponding peak table.

**Figure 5 nanomaterials-14-01086-f005:**
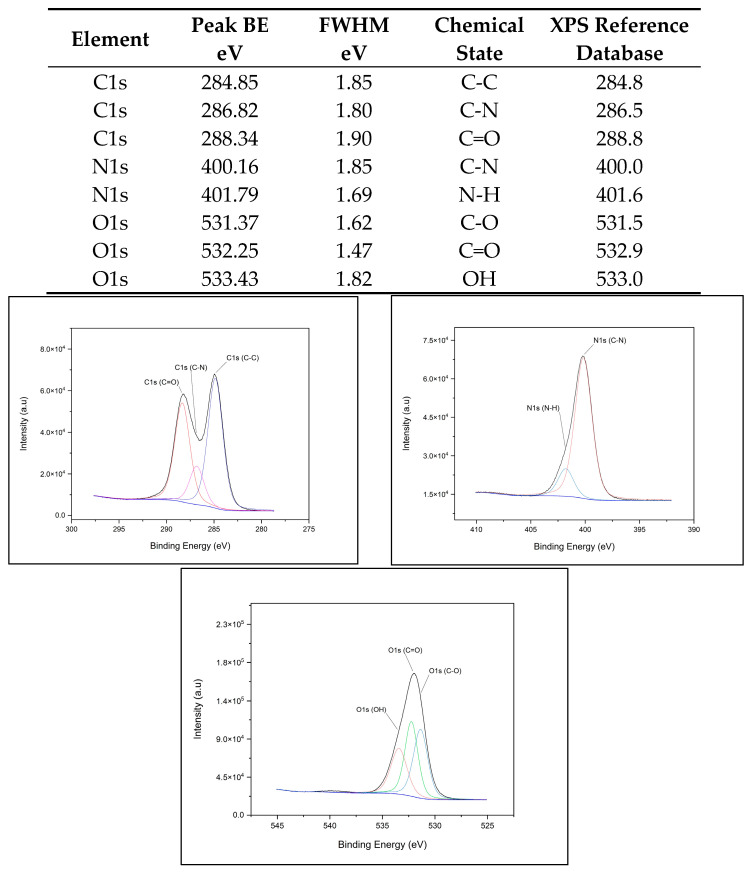
NCD QD high-resolution XPS spectra with corresponding peak table.

**Figure 6 nanomaterials-14-01086-f006:**
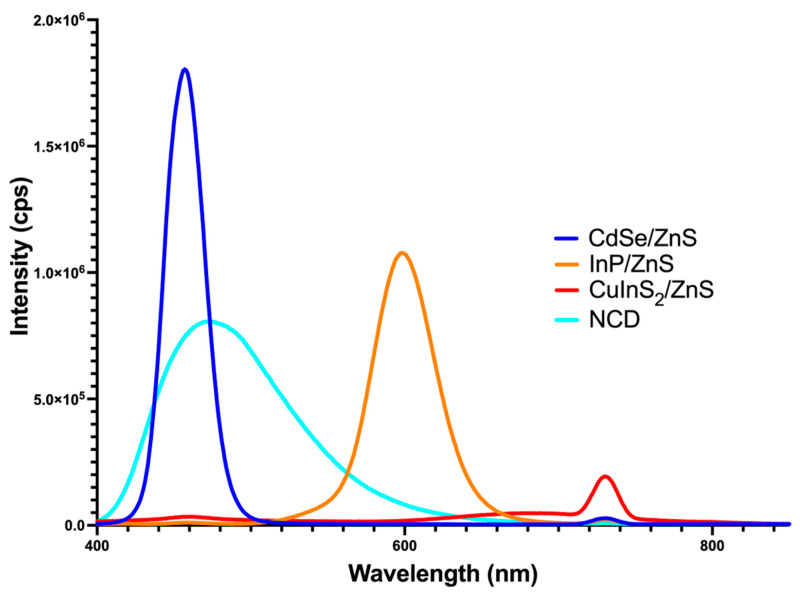
The mean fluorescent spectra for each quantum dot with lines corresponding to the color of fluorescence (n = 3). The concentrations used for reading were 50 nM for CdSe/ZnS and 100 nM for InP/ZnS, CuInS_2_/ZnS, and NCDs.

**Figure 7 nanomaterials-14-01086-f007:**
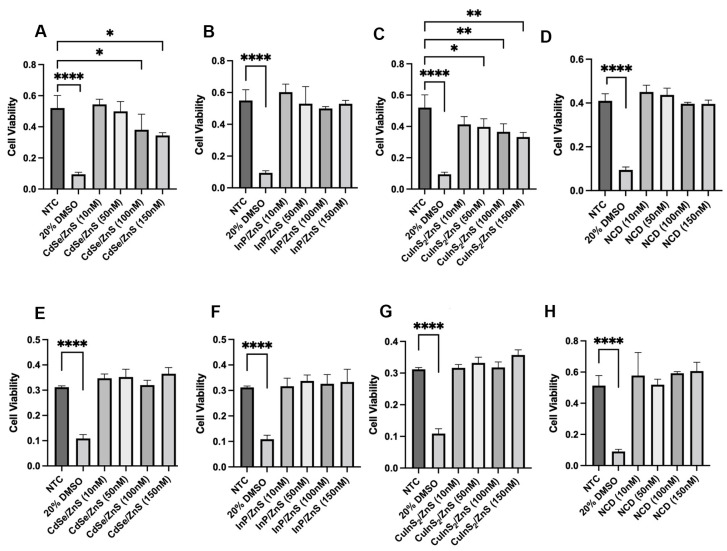
The effect of CdSe/ZnS, InP/ZnS, CuInS_2_/ZnS, and NCDs on the cell viability of HepG2 and THLE-2 liver cells compared to a non-treated control (NTC) and positive control (20% DMSO). Cell viability is determined by the average absorbance values in the delta wavelength range (450–630 nm) where a lower value correlates to less cell viability. (**A**–**D**) The top row of graphs shows the effect on cell viability in THLE-2 cells with the following order: CdSe/ZnS, InP/ZnS, CuInS_2_/ZnS, and NCDs (carbon dots). (**E**–**H**) The bottom row shows the same except in HepG2 cells. Statistical analysis is indicated by * *p* = 0.05, ** *p* = 0.01, **** *p* = 0.0001. The mean values (n = 3) include ±SEM.

**Figure 8 nanomaterials-14-01086-f008:**
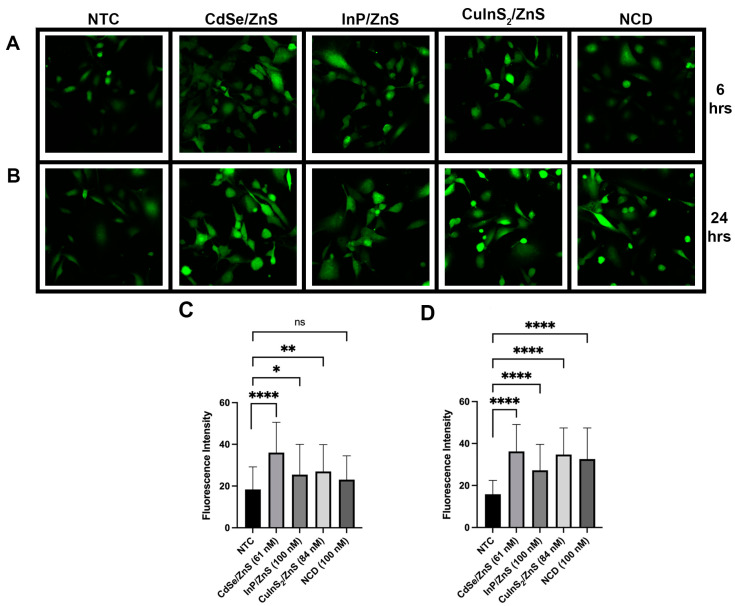
ROS images of THLE-2 cells treated with QDs. All images are representative images of a cell sample size of 60 cells repeated in triplicate (n = 3). The first row of images (**A**) is after 6 h of incubation in QDs, and the second row (**B**) is after 24 h of incubation. The figures below (**C**,**D**) represent the fluorescent intensity with ±SEM for each image quantified using ImageJ software. The figure on the bottom left (**C**) corresponds to the 6 h treatment, and the bottom right (**D**) to the 24 h treatment. Statistical analysis is indicated by * *p* = 0.05, ** *p* = 0.01, **** *p* = 0.0001.

**Figure 9 nanomaterials-14-01086-f009:**
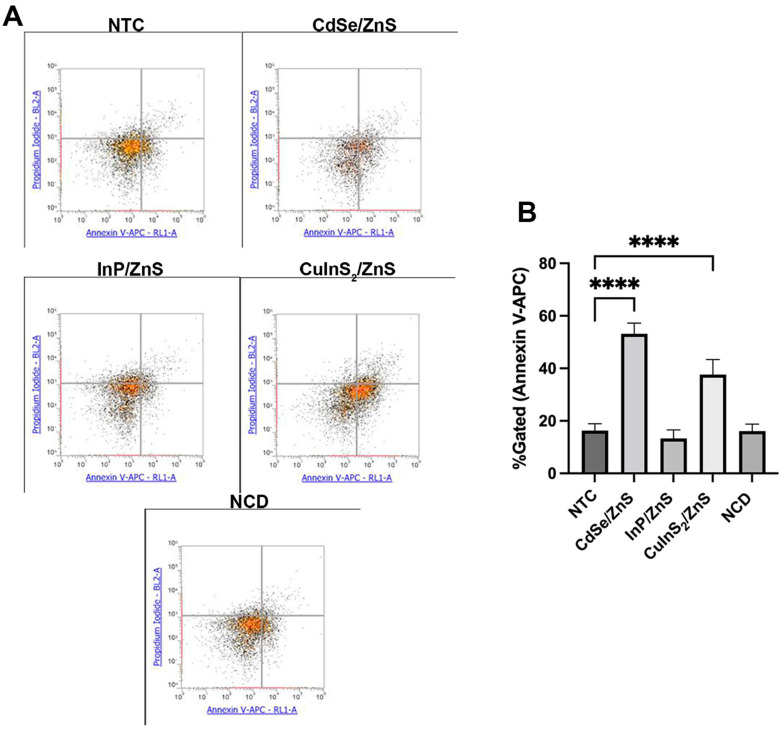
The effect of QDs on the induction of apoptosis in THLE-2. (**A**) The flow cytometry plots show Annexin V-APC (early apoptosis) on the x-axis and propidium iodide (late apoptosis) on the y-axis. (**B**) Plots the ±SEM %gated cells showing signs of early apoptosis after 24 h (n = 3). Concentrations were either IC20 values or 100 nM, consistent with other experiments. Statistical analysis is indicated by **** *p* = 0.0001.

**Figure 10 nanomaterials-14-01086-f010:**
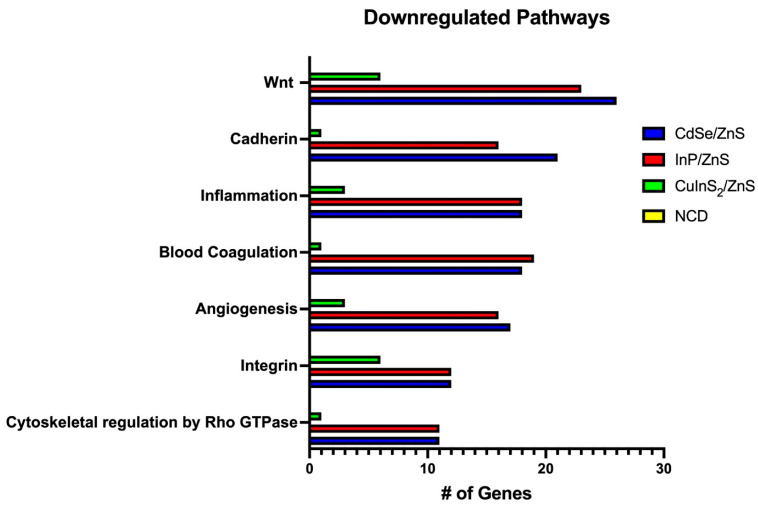
The pathways with the most genes downregulated according to Panther 18.0 functional classification analysis. Each QD is color-coded, and only differentially expressed genes were considered (fold change < −1.5, *p*-value < 0.05).

**Figure 11 nanomaterials-14-01086-f011:**
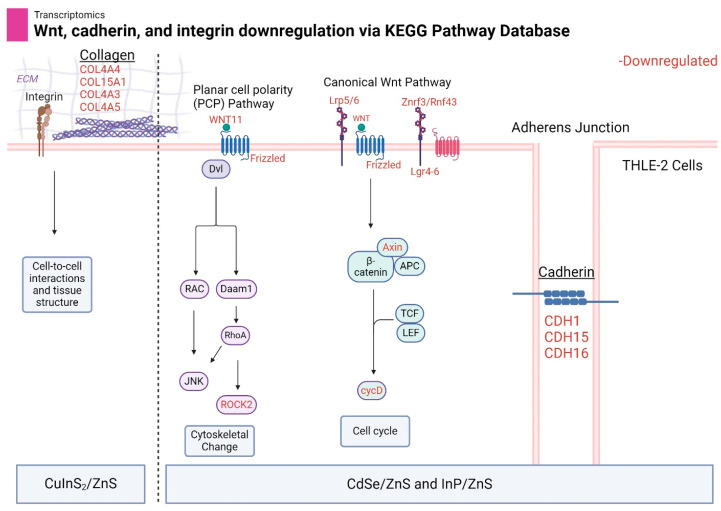
A model of adherence-related genes downregulated by CdSe/ZnS, InP/ZnS, and CuInS_2_/ZnS treatment after 24 h. Each gene in red font is considered downregulated with a fold change < −1.5 and *p*-value < 0.05. The signaling cascades were found using the KEGG database and the ShinyGo 0.80 enrichment tool. The dotted line is there to separate the primary pathways downregulated by each type of QD. Created with BioRender.com.

**Table 1 nanomaterials-14-01086-t001:** The number of upregulated and downregulated genes in QD-treated THLE-2 cells. The genes were counted if they had a significant fold change > 1.5 (upregulated) or <−1.5 (downregulated) and *p*-value < 0.05. Each treatment as listed was incubated for 24 h at their IC20 values or 100 nM if the IC20 values were not applicable.

QD Type	Number of Genes	
	Upregulated	Downregulated
CdSe/ZnS	53	1105
InP/ZnS	53	1030
CuInS_2_/ZnS	191	392
NCD	38	49

**Table 2 nanomaterials-14-01086-t002:** Differentially expressed genes shared between CdSe/ZnS, InP/ZnS, and CuInS_2_/ZnS QDs. The numbers indicate the fold change corresponding to the gene of interest. Each gene listed has a *p*-value < 0.05 compared to a non-treated control.

Gene	CdSe/ZnS	InP/ZnS	CuInS_2_/ZnS	NCD
CD82	1.706	1.713	2.089	n/a
SYT12	1.835	1.745	1.944	n/a
DUSP4	1.644	1.508	1.577	n/a

## Data Availability

All data are contained within the article.

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
