# Peer review of "A Comparison of Common Quantum Dot Alternatives to Cadmium-Based Quantum Dots on the Basis of Liver Cytotoxicity"

_nanomaterials, 2024, doi:10.3390/nano14131086_

Round 1

Reviewer 1 Report (Previous Reviewer 1)

Comments and Suggestions for Authors

After modification, the manuscript turns out to be suitable for publication.

Reviewer 2 Report (Previous Reviewer 2)

Comments and Suggestions for Authors

All my comments were addressed well, this may be accepted after the editor's review.

Comments on the Quality of English Language

Language must be improved before its acceptance.

Reviewer 3 Report (Previous Reviewer 3)

Comments and Suggestions for Authors

Authors have revised the manuscript carefully and it looks good. I recommend to publish this wok in present form.

This manuscript is a resubmission of an earlier submission. The following is a list of the peer review reports and author responses from that submission.

Round 1

Reviewer 1 Report

Comments and Suggestions for Authors

In this manuscript, the author proposed the use of some methods to monitor the effects of various quantum dots on liver cells in order to assess the toxic impacts of quantum dots on cellular function. The results are well presented and analyzed in a systematic manner with good English. Thus, the referee considers that the submitted results will give readers a valuable aid point and further opportunity for the investigations in this field. Therefore, the experimental work and the discussion presented by the authors are of fine quality & scope in order to be suitable for the publication in this journal, which is required a major revision before publication as shown below.

Comment 1:Author claimed that “Quantum dots are semiconductor nanocrystals with a core of specified elements such as cadmium”. The statement is kind of confused. Quantum dots can be synthesized with various materials i.e. Pb, P, Sc, MA, etc.. Thus, please clarify it.

Comment 2: In Figure 1, Figure3, and Figure 4C, D, the content and the numerical values on the x and y axis, are not clearly visible at a 100% scale. Please provide clear and legible figures for readers.

Comment 3:Indeed, XPS is an alternative tool for quantum dot characterization, but it’s generally limited to surface information rater than the whole quantum dot due to the transmission limitation of X-ray. Thus, some direct evidence (i.e. TEM image and elementary spectra) should be provided.

Comment 4:The PL spectra for quantum dots appeared a little strange. For instance, a broadening band of PL peak for CuInS2/ZnS quantum dots with a quite low peak intensity in comparison with InP/ZnS quantum dots. Meanwhile, double peaks appeared for the CdS/ZnS quantum dots. All those evidences suggested ununified size distribution of quantum dots. Therefore, the quality of those quantum dots should be proved, which can significantly decide the results of the experiments.

Another concern still confused the reviewer.  As the quantum efficiency for each quantum are different, as shown in PL spectra. Generally, additional amounts of quantum dots can be provided during imaging. But, same concentration of quantum dots was test for the toxicity. Authors are strongly suggested to provide a convincible explanation.

Comment 5:There are some mistakes need to be corrected. (1) The full name should be always provided first before abbreviation (i.e. XXT, ROS in Abstract). (2) In Abstract CuInS2/ZnS (38%), the number 2 should be subscript (Line 20) . (3) The title shouldn't be in all capital form.

Comments on the Quality of English Language

The language generally looks fine for reviwer.

Reviewer 2 Report

Comments and Suggestions for Authors

Harris et al. investigate the toxicity of quantum dot (QD) alternatives and examine their cellular interactions, but some adjustments are needed before the manuscript can be considered.

  1. The title could be clearer.
  2. There are inconsistencies in the use of superscripts and subscripts throughout the manuscript.
  3. The keywords are unclear and should be improved for better understanding.
  4. In Section 2.2 (third line), there is a citation labeled as number 4 in a different format. Please address this inconsistency.
  5. Clarify what "working concentration" refers to in Section 2.1.
  6. In Section 2.3, "we" is used around 10 times. Consider avoiding "we".
  7. Remove links from Section 2.4.1, as they are not allowed in the article.
  8. Provide more details in Sections 2.6.1 and 2.7.1 to improve understanding.
  9. "3.8. QDs reveal no distinguished pattern of upregulation" is not an appropriate title.
  10. Remove the phrase "Upregulated Fold Change" from Table 3.
  11. The subheading in the conclusions, "InP/ZnS and NCD show promise as potential cadmium QD alternatives," is unnecessary.
Comments on the Quality of English Language

Major revision needed.

Reviewer 3 Report

Comments and Suggestions for Authors

There are several important concerns that should be rectified before publication.

1. In this report, authors report a replacement of CdSe/ZnS due to toxicity. The importance of CdSe/ZnS is not very clear. Also, CDSe/ZnS QD are highly fluorescent and have high yield. So, It is essential to show the quantum yield of the alternative QDs.

2. In Fluorescence spectra, InP/ZnS and CuInS2/ZnS QDs show fluorescence at higher wavelengths, but there are no clear explanations in the manuscript. Excitation wavelengths are also not mentioned for any QDs.

3. It will be more apparent to readers if UV spectra are included in the manuscript.

4. In XPS analysis, high-resolution XPS spectra will be more accurate in understanding the binding energy of the components.

5. All QD formula needs to be checked carefully. Subscripts are missing.

5. Font sizes are different in some sentences, which should rectified.

6. The conclusions are not clear.

7. In table 2; Instead of #, number of genes will be more clear.

8. One general question from authors: How are new QDs better than other QDs used for vivo analysis?

9. In Abstract: a minor typo "carbon dots." I think it should be QD, not carbon dots.